# Diagnostic Accuracy of PET/CT or PET/MRI Using PSMA-Targeting Radiopharmaceuticals in High-Grade Gliomas: A Systematic Review and a Bivariate Meta-Analysis

**DOI:** 10.3390/diagnostics12071665

**Published:** 2022-07-08

**Authors:** Barbara Muoio, Domenico Albano, Francesco Dondi, Francesco Bertagna, Valentina Garibotto, Jolanta Kunikowska, Arnoldo Piccardo, Salvatore Annunziata, Vittoria Espeli, Denis Migliorini, Giorgio Treglia

**Affiliations:** 1Oncology Institute of Southern Switzerland, Ente Ospedaliero Cantonale, 6501 Bellinzona, Switzerland; bmuoio@hotmail.it (B.M.); vittoria.espeli@eoc.ch (V.E.); 2Department of Oncology, Geneva University Hospitals, 1205 Geneva, Switzerland; denis.migliorini@hcuge.ch; 3Division of Nuclear Medicine, University of Brescia and ASST Spedali Civili Brescia, 25123 Brescia, Italy; domenico.albano@unibs.it (D.A.); f.dondi@outlook.com (F.D.); francesco.bertagna@unibs.it (F.B.); 4Division of Nuclear Medicine and Molecular Imaging, Geneva University Hospitals and Geneva University, 1205 Geneva, Switzerland; valentina.garibotto@hcuge.ch; 5Nuclear Medicine Department, Medical University of Warsaw, 02-091 Warsaw, Poland; jolanta.kunikowska@wum.edu.pl; 6Department of Nuclear Medicine, Ente Ospedaliero “Ospedali Galliera”, 16128 Genoa, Italy; arnoldo.piccardo@galliera.it; 7Unit of Nuclear Medicine, TracerGLab, Department of Diagnostic Imaging, Oncological Radiotherapy and Hematology, IRCCS A. Gemelli University Polyclinic Foundation, 00168 Rome, Italy; salvatore.annunziata@policlinicogemelli.it; 8Imaging Institute of Southern Switzerland, Ente Ospedaliero Cantonale, 6501 Bellinzona, Switzerland; 9Faculty of Biology and Medicine, University of Lausanne, 1011 Lausanne, Switzerland; 10Faculty of Biomedical sciences, Università della Svizzera italiana, 6900 Lugano, Switzerland

**Keywords:** PET, positron emission tomography, nuclear medicine, PSMA, brain tumors, glioblastoma, glioma, neuro-oncology, meta-analysis

## Abstract

Background: Several studies proposed the use of positron emission tomography (PET) with Prostate Specific Membrane Antigen (PSMA)-targeting radiopharmaceuticals in brain tumors. Our aim is to calculate the diagnostic accuracy of these methods in high-grade gliomas (HGG) with a bivariate meta-analysis. Methods: A comprehensive literature search of studies on the diagnostic accuracy of PET/CT or PET/MRI with PSMA-targeting radiopharmaceuticals in HGG was performed. Original articles evaluating these imaging methods both in the differential diagnosis between HGG and low-grade gliomas (LGG) and in the assessment of suspicious HGG recurrence were included. Pooled sensitivity, specificity, positive and negative likelihood ratios (LR+ and LR-), and diagnostic odds ratio (DOR) including 95% confidence intervals (95% CI) were calculated. Statistical heterogeneity was also assessed using the I^2^ test. Results: The meta-analysis of six selected studies (157 patients) provided the following results about PET/CT or PET/MRI with PSMA-targeting radiopharmaceuticals in the diagnosis of HGG: sensitivity 98.2% (95% CI: 75.3–99.9%), specificity 91.2% (95% CI: 68.4–98.1%), LR+ 4.5 (95% CI: 2.2–9.3), LR− 0.07 (95% CI: 0.04–0.15), and DOR 70.1 (95% CI: 19.6–250.9). No significant statistical heterogeneity among the included studies was found (I^2^ = 0%). Conclusions: the quantitative data provided demonstrate the high diagnostic accuracy of PET/CT or PET/MRI with PSMA-targeting radiopharmaceuticals for HGG detection. However, more studies are needed to confirm the promising role of PSMA-targeted PET in this clinical setting.

## 1. Introduction

Gliomas are the most common central nervous system primary tumors originating from the glial cells [1]. The annual incidence of gliomas is about six per 100,000 cases worldwide [1,2]. About grading, gliomas are most often referred to as low-grade gliomas (LGG) or high-grade gliomas (HGG), based on the growth potential and aggressiveness of the tumors [3]. HGG (including glioblastomas) are still burdened by a poor prognosis and high mortality, regardless of the type of therapy; conversely, LGG present a better prognosis compared to HGG [4].

According to recent evidence-based guidelines, brain Magnetic Resonance Imaging (MRI) without and with the administration of a gadolinium-based contrast agent is the diagnostic imaging modality of choice in the evaluation of patients with gliomas [2].

Positron emission tomography (PET) is a nuclear medicine imaging method that, using different radiopharmaceuticals evaluating various metabolic pathways, can detect in advance functional changes in brain tumors, which usually occur before the development of morphological changes detected by conventional neuroimaging techniques such as computed tomography (CT) and MRI. Hybrid imaging techniques (PET/CT and PET/MRI) combining functional and morphological information may be useful methods for discriminating LGG and HGG or for detecting HGG recurrence [5,6]. In this regard, different radiopharmaceuticals have been used to evaluate HGG, including fluorine-18 fluorodeoxyglucose ([^18^F]FDG) and several radiolabelled amino acid tracers (such as [^18^F]FET, [^18^F]FDOPA and [^11^C]methionine) with good diagnostic performance according to published meta-analyses [7].

Prostate specific membrane antigen (PSMA), also known as glutamate carboxypeptidase II, is a membrane antigen overexpressed in the majority of prostate cancer cells; therefore, PSMA has proven to be a good target for diagnosis and therapy (theranostics) of prostate cancer using a variety of radiolabelled PSMA-targeting radiopharmaceuticals [8,9,10]. Undoubtedly, PSMA PET is a hot topic of imaging in the last years. Recent literature data and a growing body of evidence demonstrated that PSMA-targeted PET ameliorates staging and restaging of prostate cancer patients, changing the management in a significant percentage of cases [11,12]. Beyond prostate cancer, PSMA has been also demonstrated to be overexpressed by the neovasculature of many other solid tumors, including HGG [13,14,15,16]; this could be the rationale for using PET/CT or PET/MRI with PSMA-targeting radiopharmaceuticals in HGG or other solid tumors where [^18^F]FDG PET has demonstrated low diagnostic accuracy [16]. Compared to other PET radiopharmaceuticals used for HGG, PSMA-targeting radiopharmaceuticals may evaluate a different tumor characteristic (tumor neoangiogenesis). Interestingly, for HGG there could also be a potential advantage of a theranostic approach offered by PSMA-targeted agents, which is not possible with [^18^F]FDG and radiolabelled amino acid tracers [11,16].

Several studies have used PSMA-targeting radiopharmaceuticals for PET imaging of HGG as reported in the literature [14,15]. The aim of this work is to perform a bivariate meta-analysis to calculate the diagnostic accuracy of PET/CT or PET/MRI with PSMA-targeting radiopharmaceuticals in patients with HGG in the following clinical settings: differential diagnosis among HGG and LGG before treatment and assessment of suspicious HGG recurrence after treatment.

## 2. Materials and Methods

### 2.1. Protocol

This systematic review and meta-analysis has been conducted according to a predefined protocol [17], and the article has been written according to the “Preferred Reporting Items for a Systematic Review and Meta-Analysis of Diagnostic Test Accuracy Studies” (PRISMA-DTA statement) [18]. The complete PRISMA-DTA checklist is available as Appendix A. The protocol was not registered.

The first step of the process was the definition of a clear review question, including the index test (PET/CT or PET/MRI with PSMA-targeting radiopharmaceuticals), the patient population and target condition (patients with suspicious HGG at the diagnosis or suspicious recurrence of HGG after treatment) and the outcome measures (diagnostic accuracy measures, such as sensitivity and specificity).

### 2.2. Literature Search Strategy and Information Sources

After the definition of the review question, a comprehensive literature search has been performed independently by three authors using three electronic bibliographic databases (PubMed/MEDLINE, Embase and Cochrane library) searching for studies evaluating the diagnostic accuracy of PET/CT or PET/MRI with PSMA-targeting radiopharmaceuticals in HGG. The bibliographic databases were searched until 22 May 2022.

A predefined search algorithm based on a combination of text words (with truncation) related to the review question was used: (A) “PSMA” AND B) “glioma*” OR “glioblastoma*” OR “brain” OR “nerv*” OR “glial”. Date limits or language restrictions were not applied to the search of electronic databases. Furthermore, to achieve a more comprehensive search, the references of retrieved studies were also searched for potential additional eligible articles.

### 2.3. Eligibility Criteria

According to the predefined review question, clinical studies reporting information on the diagnostic accuracy of PSMA-targeted PET/CT or PET/MRI both in the differential diagnosis among HGG and LGG and in the assessment of suspicious HGG recurrence were eligible for inclusion in the systematic review and meta-analysis. Exclusion criteria for the systematic review (qualitative analysis) were: review articles/letters/comments/editorials in the topic of interest; case reports/small case series (less than 5 patients) in the topic of interest; articles outside the field of interest (including preclinical studies or studies not using positron emitting radionuclides). Additional exclusion criteria for the meta-analysis (quantitative analysis) were: articles not providing sufficient information to reassess the sensitivity or specificity of PSMA-targeted PET/CT or PET/MRI (when true positive, false positive, true negative and false negative findings were not reported); articles with possible patient data overlap with another study (in this case, all the selected articles were included in the systematic review, whereas only the article with the most complete information was included in the meta-analysis).

### 2.4. Study Selection

Titles and abstracts of the records obtained by using the predefined literature search strategy and information sources were independently screened by three reviewers based on the predefined inclusion and exclusion criteria mentioned above. The full text of selected original articles was independently screened to assess for their final inclusion for both the systematic review and the meta-analysis. For all the screened records using the bibliographic databases, the reviewers provided a final decision on inclusion or exclusion in the review specifying the reason. Disagreements among the reviewers were solved through an online consensus meeting.

### 2.5. Data Collection Process and Data Extraction

The data collection process was independently carried out by three reviewers to minimize possible bias. Data extraction from reports (using full text, tables and/or figures) was performed using piloted forms. In selected cases, for obtaining and confirming data from investigators, reviewers contacted corresponding authors by e-mail. For each study eligible for the systematic review, the data extracted included: general study information (authors, year of publication, country, study design, funding sources); patient characteristics (sample size, age, sex ratio, type of glial tumor, clinical setting and prior imaging testing); index text characteristics (type of PSMA-targeting radiopharmaceutical, type of hybrid imaging method, patient preparation protocol, radiopharmaceutical injected activity, time interval between PSMA-targeting tracer injection and image acquisition, protocol for the image analysis); data on the diagnostic accuracy of PSMA-targeted PET/CT or PET/MRI in HGG on a per-patient-based analysis (including true positive, true negative, false positive and false negative findings, sensitivity, specificity, positive and negative predictive values, diagnostic accuracy); and type of reference standard used. Any discrepancies among the reviewers about data extraction were solved by consensus.

### 2.6. Quality Assessment

The selected method used for assessing risk of bias in individual studies and concerns regarding the applicability to the review question was QUADAS-2, a tool for assessing quality in diagnostic test accuracy studies [19]. The quality of the included studies in the systematic review and meta-analysis was independently assessed by three reviewers. Four domains (patient selection, index test, reference standard, and flow and timing) were assessed in terms of risk of bias, and three domains were assessed in terms of concerns regarding applicability (patient selection, index test and reference standard). Any discrepancies among the reviewers about the quality assessment were solved by consensus.

### 2.7. Statistical Analysis and Diagnostic Accuracy Measures

Diagnostic accuracy measures were calculated on a per-patient-based analysis, taking into account data extracted from each study (true positive, false positive, true negative, and false negative findings). Pooled sensitivity and specificity were calculated using a bivariate random-effects model for statistical pooling of data. For the diagnostic test accuracy meta-analysis, the bivariate analysis is preferred compared to univariate analysis because it considers any possible correlation between sensitivity and specificity [17]. Beyond pooled sensitivity and specificity, further diagnostic measures were calculated using a random-effects model, including positive and negative likelihood ratios (LR+ and LR-) as well as diagnostic odds ratio (DOR). Pooled data were provided with 95% confidence intervals values (95% CI) and displayed using forest plots. To summarize the diagnostic performance of the index test, a summary receiver operating characteristic (SROC) curve was also provided. The SROC curve is created by plotting the true positive rate (sensitivity) against the false positive rate (1-sensitivity) at various threshold settings [17]. Statistical heterogeneity was assessed using the I-square index (I^2^), with significant heterogeneity for values >50% [17]. OpenMeta[Analyst]^®^ software funded by the Agency for Healthcare Research and Quality (AHRQ) (Rockville, MD, USA) was used for the statistical analyses.

### 2.8. Additional Analyses

Subgroup analyses taking into account basic study and patient characteristics as well as technical aspects or clinical settings were planned in case of significant statistical heterogeneity.

## 3. Results

### 3.1. Literature Search and Study Selection

Overall, 209 records were identified and screened through the comprehensive literature search described above. Taking into account predefined eligibility criteria, these 209 records were assessed for eligibility and 201 records were excluded (178 as not in the field of interest, 6 as reviews, editorials, or letters, and 17 as case reports); 8 remaining articles were judged as eligible for inclusion in the systematic review (qualitative synthesis) after full-text assessment [20,21,22,23,24,25,26,27]. No additional studies were eligible for inclusion after screening the references of these articles. Six out of eight articles were included in the meta-analysis (quantitative synthesis) [20,21,22,23,25,26]; two studies included in the systematic review were excluded from the meta-analysis [24,27] due to possible partial patient data overlap with other studies of the same research group. Figure 1 summarizes the study selection process.

### 3.2. Study Characteristics

The characteristics of the eight studies eligible for the systematic review (qualitative analysis) including 178 patients with gliomas are presented in Table 1, Table 2 and Table 3. About general study information (Table 1), these studies were published in the last six years (between 2017 and 2022). Several countries from Europe and Asia were represented. Three studies (37.5%) were prospective, one (12.5%) was retrospective, and in four articles (50%) the study design was not declared. All except two were single centre studies (75%). In most of the studies, no funding was declared.

Regarding the patient key characteristics (Table 2), the sample size ranged from 6 to 35 patients with gliomas. Mean and median age of patients ranged from 37 to 59.5 years. About the sex ratio, the percentage of male patients ranged from 43% to 80%. Among all included patients with glioma, 146 (88%) had HGG, and the remaining 32 patients (12%) had LGG. Most of the HGG were glioblastomas (115 out of 146; 79%). About the clinical setting (Table 2), the index test was performed in patients with glioma for initial diagnosis to differentiate HGG and LGG (*n* = 3 studies), to evaluate a suspicious recurrence of HGG after treatment (*n* = 3 studies), or for both indications (*n* = 2 studies). Prior imaging testing included MRI in all the included studies and additional [^18^F]FDG PET/CT in three of them. Contrast-enhanced MRI was used for comparison in all the included studies. Only some studies reported the use of functional MRI sequences [20,21,24,26].

Regarding the index test key characteristics (Table 3), heterogeneous aspects among the included studies were found. The radiopharmaceutical injected was [^68^Ga]Ga-PSMA-11 in most of the cases.

Seven studies (87.5%) used hybrid PET/CT, whereas only one study used a PET/MRI tomograph. PET/CT was performed using a low-dose CT acquisition for attenuation correction and anatomical localization. PET was fused with previous MRI in some studies.

The radiopharmaceutical-injected activity ranged between 100 and 185 MBq (in absolute values) and between 1.8 and 2.2 MBq/kg. The time interval between radiopharmaceutical injection and PET image acquisition ranged from 45 to 60 min. The PET image analysis was performed by using qualitative (visual) analysis and additional semi-quantitative analysis in all the studies. Additional semi-quantitative analyses included the calculation of the maximal and mean standardized uptake values (SUV_max_ and SUV_mean_) of the detected lesions, measured using spherical volume of interest (VOI). Target-to-background uptake ratios (TBR) were also frequently calculated using SUV_max_ of the lesion divided by SUV_max_ of the background. Background uptake was defined as contralateral normal cerebral uptake or contralateral cerebellar uptake. Less frequently other semi-quantitative measures were used, including PSMA-tumor volume (TV), tumor-to-liver uptake ratio (TLR), and SUV_peak_.

### 3.3. Risk of Bias and Applicability

The overall evaluation of risk of bias and concerns regarding applicability for studies included in the systematic review according to QUADAS-2 is presented in Figure 2.

### 3.4. Results of Individual Studies (Qualitative Synthesis)

Diagnostic accuracy data of PET/CT or PET/MRI with PSMA-targeting radiopharmaceuticals in HGG patients for each study are listed in Table 4. Overall, the index test has demonstrated an excellent diagnostic performance in detecting HGG in all studies included in the systematic review, both on a per patient- and on a per lesion-based analysis and in different clinical settings (at initial diagnosis and in the suspicious recurrence after treatment) [20,21,22,23,24,25,26,27]. Moreover, PET/CT or PET/MRI with PSMA-targeting radiopharmaceuticals was very useful in detecting multifocal disease in HGG patients [20,21,23].

Regarding toxicity and safety, the injection of PSMA-targeting radiopharmaceuticals was well tolerated, without any adverse event recorded [20,24]. The image quality of PET with PSMA-targeting radiopharmaceuticals was judged adequate [22] and interpretation of the images was very easy due to the absence of physiological radiopharmaceutical uptake in the normal brain parenchyma, without ambiguity in deciding whether the scan was positive or negative [26]. When reported, interobserver concordance for abnormal sites of radiopharmaceutical uptake at PSMA-targeted PET imaging was excellent [21,26].

In the setting of initial diagnosis of gliomas, compared to LGG, HGG are usually characterized by increased PSMA-targeting radiopharmaceutical uptake [22,23]. Average TBR in HGG ranged from 11.52 to 13.95 and average TBR in LGG ranged from 1.29 to 3.42 [22,25]. Tumor grade and proliferation index (Ki-67) in patients with glioma were found to have a correlation with uptake of PSMA-targeting radiopharmaceuticals [22,23,25]. The index test was more sensitive than MRI in distinguishing HGG from LGG, even if there was no statistically significant difference among their specificities [23].

In the setting of suspicious recurrence of HGG, no significant difference in SUV was demonstrated among Grade III and Grade IV HGG recurrence [20,21], whereas a significant difference of radiopharmaceutical uptake was demonstrated among HGG recurrence and radiation necrosis, which did not show significant radiopharmaceutical uptake [20,21]. Concordance between MRI and PET with PSMA-targeting radiopharmaceuticals was high for patient-wise and lesion-wise detection in recurrent HGG: uptake of PSMA-targeting radiopharmaceuticals was found precisely in the tumor regions that showed contrast enhancement on MRI, suggesting that blood-brain barrier (BBB) damage could be a common factor influencing both contrast enhancement on MRI and PSMA-targeting radiopharmaceutical uptake on PET [20,21].

A strong concordance between [^18^F]FDG PET/CT and [^68^Ga]Ga-PSMA PET/CT findings was demonstrated in the initial diagnosis and in evaluation of suspicious recurrence of HGG [22,25,27]; however, PET with PSMA-targeting radiopharmaceuticals was more accurate than [^18^F]FDG PET, allowing for better discrimination among HGG and LGG and for better evaluation of the presence or absence of HGG recurrence, due to the absence of physiological radiopharmaceutical uptake in normal brain parenchyma (which is evident on [^18^F]FDG PET) [22,25,27]. Liu et al. reported that average TBR for HGG and LGG were 11.52 and 1.29, respectively, on PSMA-targeted PET and 1.21 and 0.88, respectively, on [^18^F]FDG PET [22].

Only one study has correlated the in vivo and in vitro PSMA expression in gliomas by using PSMA-targeted PET and immunohistochemistry staining for PSMA, respectively [22]. The expression of PSMA in the resected glioma tissues was confirmed to be positive in some of HGG tumors and no or only low PSMA expression for LGG has been reported (0/14 = 0% grade II; 2/4 = 50% grade III; and 9/12 = 75% grade IV) [22].

### 3.5. Meta-Analysis (Quantitative Synthesis)

Six studies including 157 patients with glioma were selected for the bivariate patient-based meta-analysis [20,21,22,23,25,26].

The sensitivity of PET/CT or PET/MRI with PSMA-targeting radiopharmaceuticals for detecting HGG ranged from 85.7% to 100%, with a pooled estimate of 98.2% (95% CI: 75.3–99.9%).

The specificity of PET/CT or PET/MRI with PSMA-targeting radiopharmaceuticals for detecting HGG ranged from 66.7% to 100%, with a pooled estimate of 91.2% (95% CI: 68.4–98.1%). A summary ROC curve is shown in Figure 3.

The pooled LR+, LR- and DOR were 4.5 (95% CI: 2.2–9.3), 0.07 (95% CI: 0.04–0.15) and 70.1 (95% CI: 19.6–250.9), respectively (Figure 4, Figure 5 and Figure 6).

No significant statistical heterogeneity among the included studies was found for all the metrics evaluated (I^2^ = 0%).

Subgroup analyses did not show a significant difference in the diagnostic accuracy of the index test in different clinical settings (initial diagnosis of HGG and suspicious recurrence of HGG). For differentiating among HGG and LGG, the pooled sensitivity and specificity of PSMA-targeted PET was 86.5% (95% CI: 73–94%) and 89.2% (95% CI: 71–96.5%), respectively. For diagnosis of HGG recurrence after treatment, the pooled sensitivity and specificity of PSMA-targeted PET was 98.5% (95% CI: 90–99.8%) and 77.5% (95% CI: 55–97.3%), respectively.

## 4. Discussion

A growing body of literature reported remarkable results for PSMA-targeted imaging and therapy in prostate cancer [8,9,10,11,12]. To date, most clinical research on PSMA-targeted imaging and therapy focuses on prostate cancer due to the high level of PSMA expression by prostate cancer tumor cells [28]. However, PSMA expression is not specific for prostate cancer cells, as PSMA is also expressed by neovascular endothelial cells of various malignant tumors, including HGG [28]. The first reports on immunohistochemical staining in HGG described PSMA expression only in the neovasculature and not in the tumor cells, reporting that 72% of HGG express PSMA in the neovasculature [28]. Overall, the demonstrated PSMA expression in HGG is the rationale for using PSMA-targeting imaging and therapy in these tumors.

Recent studies have assessed the diagnostic accuracy of PET/CT or PET/MRI with PSMA-targeting radiopharmaceuticals, in terms of sensitivity or specificity, for detecting HGG before treatment or in suspected recurrence after treatment [20,21,22,23,24,25,26,27]. We have pooled published data through a meta-analysis, increasing the statistical power to obtain more robust estimates of the selected outcome measures compared to the original studies. The use of a bivariate random-effects model is considered an appropriate tool for pooling sensitivity and specificity from multiple diagnostic test accuracy studies because this hierarchical statistical model considers any correlation that may exist between sensitivity and specificity [17].

Overall, despite the relatively limited data available from the literature, an excellent diagnostic accuracy of PET/CT or PET/MRI with PSMA-targeting radiopharmaceuticals for HGG detection has been demonstrated in both of the clinical settings evaluated (before and after treatment). These findings can be explained by the high expression of PSMA in the neovasculature of HGG compared to LGG or post-treatment abnormalities [11,12,13]. In particular, the differentiation between HGG recurrence and radiation necrosis is still a hurdle in the clinic. Radiation necrosis typically results in endothelial cell damage and small-vessel injury, and since PSMA is expressed on the neovasculature of HGG, maybe PSMA-targeted PET would be able to make a differentiation. No significant radiopharmaceutical uptake has been reported in cases of radiation necrosis in the articles included in our systematic review [20,21]. A single case of an increased PSMA-targeting radiopharmaceutical uptake in radiation necrosis has been published in the literature [29] but not included in our systematic review (as case reports should be excluded in systematic reviews). The authors of this case report hypothesized that uptake of PSMA-targeting radiopharmaceuticals may be nonspecific for HGG in the context of BBB breakdown from other causes and that PSMA-targeting radiotracer accumulation in nonneoplastic tissue of the brain is likely mediated by radiotracer binding to PSMA on astrocytes [29]. Overall, more data are needed to demonstrate that radiation necrosis could be a potential false positive finding of PSMA-targeted PET in the evaluation of suspicious HGG recurrence.

Furthermore, no significant uptake of PSMA-targeting radiopharmaceuticals is evident in the normal brain parenchyma (maybe because these tracers are unable to pass through the BBB or because the target is on the neovasculature of HGG and not on non-tumoral vasculature), and this facilitates the detection of brain lesions with increased PSMA expression by using this PET imaging method. Whereas HGG showed both in vivo and in vitro high expression of PSMA, tumor samples in LGG showed no or only low PSMA-expression, thus confirming the specificity of PSMA-targeting radiopharmaceuticals for HGG detection [22].

PET with PSMA-targeting radiopharmaceuticals was more sensitive than MRI in distinguishing HGG from LGG, even if there was no statistically significant difference among their specificities [23]. Concordance between MRI and PET with PSMA-targeting radiopharmaceuticals was high for detection of recurrent HGG [20,21]; these findings can be explained by similar patterns of contrast enhancement and regional cerebral blood flow on MRI and PSMA-targeting radiopharmaceuticals uptake on PET in HGG [20,21].

Overall, MRI remains the gold standard imaging method in the evaluation of glial tumors [5,6], but PET with PSMA targeting radiopharmaceuticals could be promising as a complementary imaging tool when MRI is doubtful. However, further studies should assess the clear diagnostic advantage of PSMA-targeted PET over MRI in HGG.

A clear advantage of PET with PSMA-targeting radiopharmaceuticals compared to [^18^F]FDG PET for HGG detection has been demonstrated [22,25,27]. A downside of [^18^F]FDG is indeed the very high physiological uptake in the normal brain, which negatively influences the contrast between tumoral tissue and normal brain tissue [22,25,27]. Conversely, the very low PSMA expression in normal brain parenchyma [30] and the higher TBR makes the visualization of HGG on PSMA-targeted PET easier compared to [^18^F]FDG PET. [^18^F]FDG has also been shown to be taken up in inflammatory lesions, whereas this seems to be not an issue with PSMA-targeted PET.

There are currently no studies comparing for HGG imaging PSMA-targeting radiopharmaceuticals with radiolabelled amino acids, as is the case with [^18^F]FET, [^11^C]methionine, or [^18^F]FDOPA. Radiolabelled amino acids are radiopharmaceuticals which have demonstrated their usefulness in patients with HGG [5,6]. However, our meta-analysis demonstrates that the diagnostic accuracy of PET with PSMA targeting radiopharmaceuticals for HGG detection is not inferior to that of PET with radiolabelled amino acids in the same setting, as reported in published meta-analyses [7]. For differentiating between HGG and LGG, reported pooled sensitivity and specificity were 80% and 72% for [^11^C]methionine PET, 80% and 82% for [^11^F]FET PET and 88% and 73% for ^18^F-FDOPA PET [7]. For detection of HGG recurrence after treatment, reported pooled sensitivity and specificity were 85% and 83% for [^11^C]methionine PET, 83% and 81% for [^11^F]FET PET and 92% and 76% for ^18^F-FDOPA PET [7].

The most intriguing perspective of PSMA expression in HGG lies in the concept of PSMA-targeted theranostics, considering PSMA both a diagnostic and a therapeutic target [9,11]. PSMA-radioligand therapy with alpha- or beta-emitting radionuclides provided satisfactory results in patients with metastatic castration-resistant prostate cancer [9,11]. The potential theranostic role of PSMA-targeting radiopharmaceuticals in HGG could be the true added value compared to the radiolabelled amino acids [28,31]. Since PSMA-targeted therapy demonstrated therapeutic efficacy in prostate cancer patients, maybe this treatment could also provide beneficial effects in other cancers characterized by PSMA overexpression (in tumor cells or in the neovasculature) [28]. Some case reports demonstrated promising results for the implementation of PSMA-targeted therapy in HGG [32,33]. A preclinical study has recently demonstrated that although PSMA-targeted PET in a specific murine model of glioblastoma is feasible and resulted in high TBR, absolute tumoral uptake values remained low limiting the applicability of this murine model for PSMA-targeted therapy [34]. Overall, further preclinical investigations are warranted to identify suitable models for preclinical evaluation of PSMA-targeted theranostic approaches in HGG, and further clinical studies using PSMA-targeted agents radiolabelled with alpha- or beta-emitting radionuclides are needed to confirm the value of PSMA-targeted theranostics in HGG. In addition, the dosimetric aspects related to PSMA-targeting PET tracers should be better understood, as they are essential for a theranostic approach. Furthermore, concerns about the radiotoxicity and crossfire effect of beta-emitters in the brain should be better evaluated.

Based on the literature data available so far, additional studies on the diagnostic accuracy of PSMA-targeting radiopharmaceuticals in HGG are required, in particular, more prospective and multicenter studies including a larger sample size. Furthermore, studies evaluating the impact of the index test on the management of HGG and cost-effectiveness analyses (comparing a diagnostic approach with or without the index test) will better clarify the role of PET/CT or PET/MRI with PSMA-targeting radiopharmaceuticals in HGG.

Some limitations and biases of our meta-analysis should be taken into account because they hampered obtaining definitive conclusions on this topic. First of all, a limited number of studies were available for the meta-analysis. Moreover, as a composite reference standard was used in some studies, a possible verification bias could not be excluded. Furthermore, based on the information provided in the studies selected for the meta-analysis, a selection bias could be present (due to exclusion of other tumors than gliomas) and a reporting bias (i.e., publication bias) cannot be excluded. Lastly, we did not perform a lesion-based meta-analysis, due to missing data, to reassess lesion-based sensitivity and specificity in all the included studies. Overall, we would like to suggest updating this evidence-based analysis in the future when more original articles on the selected topic will be published to obtain a more accurate analysis of the diagnostic performance of the index test.

Most of the included studies used [^68^Ga]Ga-PSMA-11 as PSMA-targeting radiopharmaceutical [20,21,23,24,25,26,27], whereas one study only used [^68^Ga]Ga-PSMA-617 [22], and this can be a potential bias in the meta-analysis, as the comparison of studies using the same PSMA-targeting radiopharmaceutical might increase confidence. However, even if PSMA-617 has been reported to have slower tumor accumulation and clearance kinetics than PSMA-11 [35], a significant difference in the detection of HGG using different PSMA-targeting radiopharmaceuticals was not demonstrated. Studies comparing different PSMA-targeting radiopharmaceuticals in detecting HGG are warranted.

Heterogeneity among studies (i.e., due to differences among patients included, methodological aspects, characteristics of the index test, study design and quality) may represent a potential source of bias in a meta-analysis [17]. Nevertheless, we have not detected a statistically significant heterogeneity among the included studies in our meta-analysis. This means that PET with PSMA-targeting radiopharmaceuticals is accurate in both clinical settings evaluated (initial diagnosis of HGG or diagnosis of HGG recurrence). About the hybrid imaging modality used, we recognize that PET/MRI was evaluated in only one included article, whereas the remaining studies evaluated PET/CT as a hybrid imaging modality; however, we do not expect a significant difference of diagnostic accuracy among PET/CT and PET/MRI with PSMA-targeting radiopharmaceuticals in HGG, considering that all patients who underwent PSMA-targeted PET/CT had a previous recent MRI for correlation or fusion.

## 5. Conclusions

The quantitative data provided by this meta-analysis demonstrate the high diagnostic accuracy and promising role of PET/CT or PET/MRI with PSMA-targeting radiopharmaceuticals for HGG detection. However, more studies are needed in this setting to confirm these findings and to clarify whether a clear advantage of PET/CT or PET/MRI with PSMA-targeting radiopharmaceuticals compared to current reference imaging methods in HGG exists and to establish clinical recommendations.

## Figures and Tables

**Figure 1 diagnostics-12-01665-f001:**
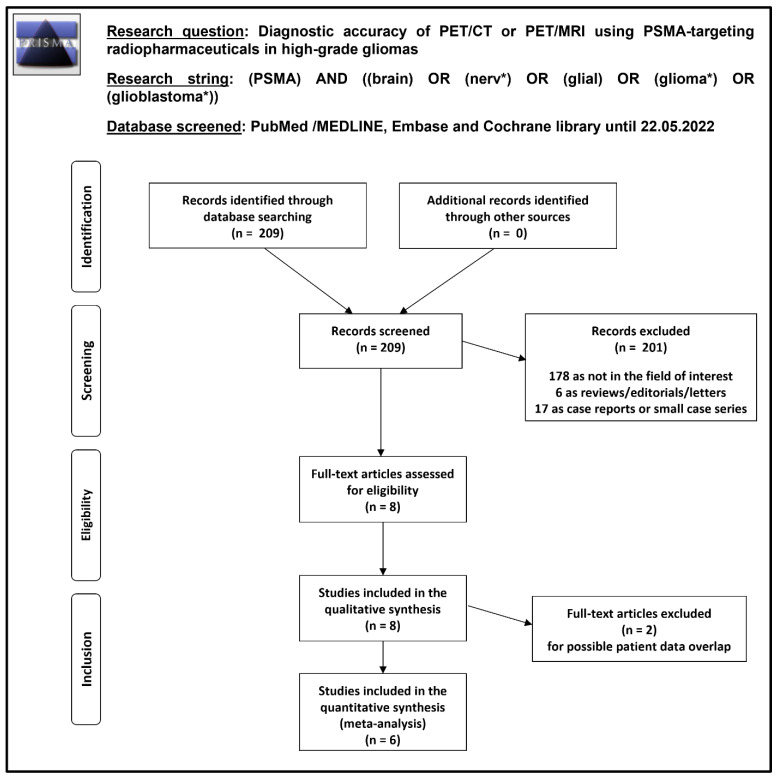
Summary of study selection process for the systematic review and meta-analysis.

**Figure 2 diagnostics-12-01665-f002:**
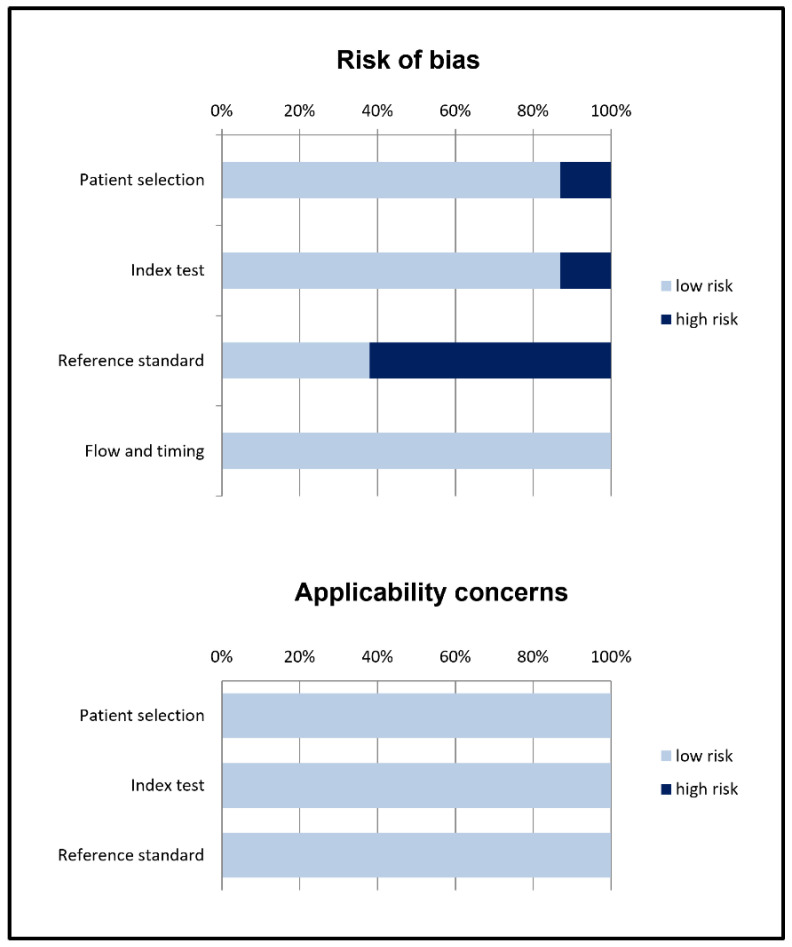
Summary of quality assessment according to QUADAS-2 tool. Studies included in the systematic review are classified as low-risk or high-risk of bias or applicability concerns for different domains (reported in the vertical axis). The horizontal axis indicates the percentage of studies. The graph indicates that for the reference standard more than half of the studies show potential high risk of bias.

**Figure 3 diagnostics-12-01665-f003:**
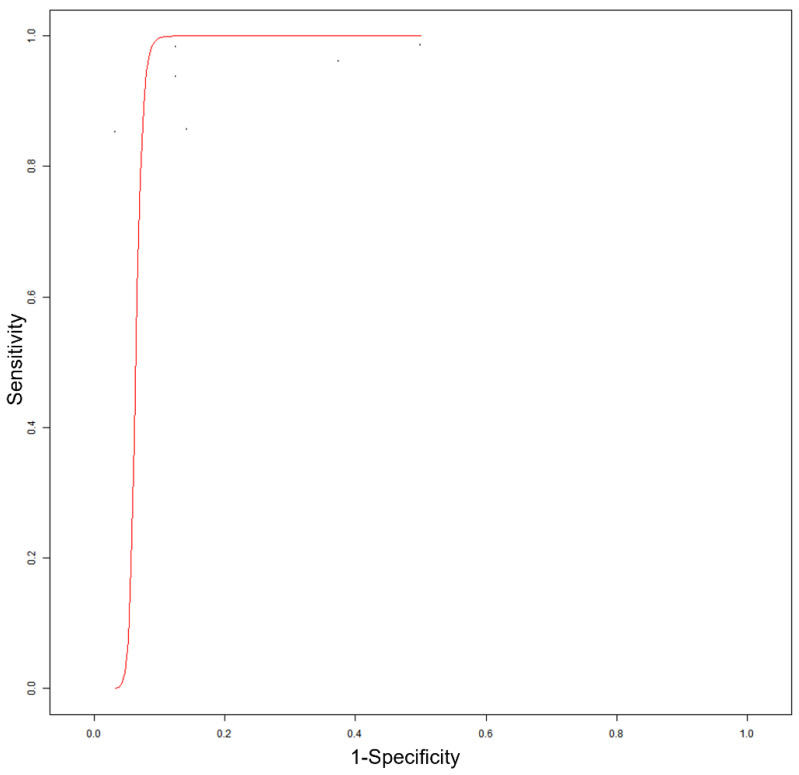
Summary ROC curve of diagnostic accuracy of the index test in high-grade glioma.

**Figure 4 diagnostics-12-01665-f004:**
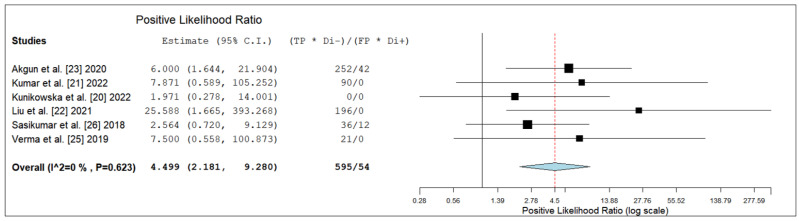
Positive likelihood ratio of the index test in high-grade glioma. Legend: 95% C.I. = 95% confidence interval; TP = true positive; TN = true negative; FP = false positive; FN = false negative.

**Figure 5 diagnostics-12-01665-f005:**
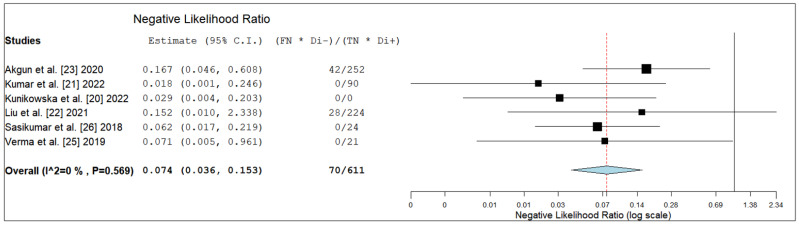
Negative likelihood ratio of the index test in high-grade glioma. Legend: 95% C.I. = 95% confidence interval; TP = true positive; TN = true negative; FP = false positive; FN = false negative.

**Figure 6 diagnostics-12-01665-f006:**
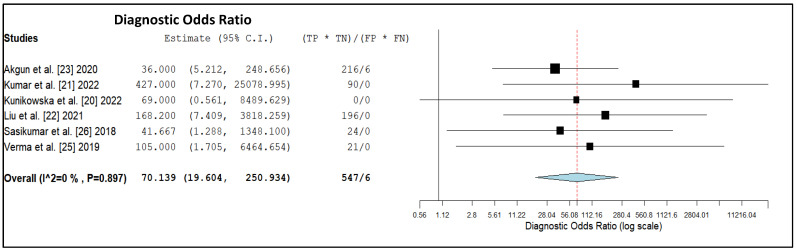
Diagnostic Odds ratio of the index test in high-grade glioma. Legend: 95% C.I. = 95% confidence interval; TP = true positive; TN = true negative; FP = false positive; FN = false negative.

**Table 1 diagnostics-12-01665-t001:** General study information.

Authors [Ref.]	Year	Country	Study Design/Number of Centers Involved	Funding Sources
Akgun et al. [23]	2020	Turkey	Prospective/bicentric	None declared
Kumar et al. [21]	2022	India	Prospective/monocentric	None declared
Kunikowska et al. [20]	2022	Poland	Not reported/monocentric	None declared
Kunikowska et al. [24]	2020	Poland	Not reported/monocentric	None declared
Liu et al. [22]	2021	China	Retrospective/monocentric	Natural Science Foundation of China
Sasikumar et al. [26]	2018	India	Prospective/bicentric	None declared
Sasikumar et al. [27]	2017	India	Not reported/monocentric	None declared
Verma et al. [25]	2019	India	Not reported/monocentric	None declared

**Table 2 diagnostics-12-01665-t002:** Patient key characteristics and clinical setting.

Authors [Ref.]	Sample Size (Gliomas)	Mean/Median Age (Years)	Male %	Type of Glioma (Grade II/III/IV)	Clinical Setting	Prior Imaging
Akgun et al. [23]	35	mean: 59.5	49%	14/6/15	HGG vs. LGG	MRI
Kumar et al. [21]	33	median: 37	67%	0/12/21	suspicious recurrence of HGG	MRI
Kunikowska et al. [20]	34	mean: 44.5	65%	0/6/28	suspicious recurrence of HGG	MRI
Kunikowska et al. [24]	15	mean: 44	67%	0/0/15	suspicious recurrence of HGG	MRI
Liu et al. [22]	30	mean: 50	43%	14/4/12	HGG vs. LGG	MRI and [^18^F]FDG PET/CT
Sasikumar et al. [26]	15	median: 50	80%	1/3/11	initial diagnosis or suspicious recurrence of HGG	MRI
Sasikumar et al. [27]	6	mean: 40	60%	0/0/6	initial diagnosis or suspicious recurrence of HGG	MRI and [^18^F]FDG PET/CT
Verma et al. [25]	10	mean: 52	80%	3/0/7	HGG vs. LGG	MRI and [^18^F]FDG PET/CT

Legend: HGG = high grade gliomas (grade III and IV); LGG = low grade gliomas (grade II); MRI = magnetic resonance imaging; [^18^F]FDG PET/CT: fluorine-18 fluorodeoxyglucose positron emission tomography/computed tomography.

**Table 3 diagnostics-12-01665-t003:** Index test key characteristics.

Authors [Ref.]	Tracer	Hybrid Imaging	Tomograph	Injected Activity	Time from Injection to Acquisition (Minutes)	Image Analysis
Akgun et al. [23]	[^68^Ga]Ga-PSMA-11	PET/MRI	SIGNA (GE)	150.6 ± 31.8 MBq	57.5 ± 3.53	visual and semi-quantitative (SUV_max_, SUV_peak_, SUV_mean_)
Kumar et al. [21]	[^68^Ga]Ga-PSMA-11	PET/CT + fusion with MRI	Biograph mCT (Siemens)	148–185 MBq	60	visual and semi-quantitative (SUV_max_, SUV_mean_, TBR, TV)
Kunikowska et al. [20]	[^68^Ga]Ga-PSMA-11	PET/CT + fusion with MRI	Biograph 64 TruePoint (Siemens)	2 MBq/kg	60	visual and semi-quantitative (SUV_max_, SUV_mean_, TBR, TLR, TV)
Kunikowska et al. [24]	[^68^Ga]Ga-PSMA-11	PET/CT + fusion with MRI	Biograph 64 TruePoint (Siemens)	2 MBq/kg	60	visual and semi-quantitative (SUV_max_, SUV_mean_, TBR, TLR, TV)
Liu et al. [22]	[^68^Ga]Ga-PSMA-617	PET/CT + correlation with MRI	Biograph 40 (Siemens)	1.8–2.2 MBq/kg	60	visual and semi-quantitative (SUV_max_, SUV_mean_, TBR)
Sasikumar et al. [26]	[^68^Ga]Ga-PSMA-11	PET/CT	Biograph 6 TruePoint (Siemens) or Gemini GLX (Philips)	NR	45–60	visual and semi-quantitative (SUV_max_, TBR)
Sasikumar et al. [27]	[^68^Ga]Ga-PSMA-11	PET/CT	Biograph 6 TruePoint (Siemens)	100 ± 19 MBq	60	visual and semi-quantitative (SUV_max_, TBR)
Verma et al. [25]	[^68^Ga]Ga-PSMA-11	PET/CT	Gemini TOF (Philips)	NR	NR	visual and semi-quantitative (SUV_max_ and TBR)

Legend: CT = computed tomography; MRI = magnetic resonance imaging; NR = not reported; PET = positron emission tomography; PSMA = prostate specific membrane antigen; SUV = standardized uptake value; TBR = tumor-to-background ratio; TLR = target-to-liver ratio; TV = tumor volume.

**Table 4 diagnostics-12-01665-t004:** Diagnostic accuracy data of the index test in high-grade gliomas on a patient-based analysis.

Authors [Ref.]	Reference Standard	TP	FP	TN	FN	Sen	Spe	PPV	NPV	Acc
Akgun et al. [23]	Histology	18	2	12	3	85.7%	85.7%	90%	80%	85.7%
Kumar et al. [21]	Histology or clinical/imaging FU	30	0	3	0	100%	100%	100%	100%	100%
Kunikowska et al. [20]	Histology or clinical/imaging FU	34	0	0	0	100%	NC	100%	NC	100%
Kunikowska et al. [24] *	Histology or clinical/imaging FU	15	0	0	0	100%	NC	100%	NC	100%
Liu et al. [22]	Histology	14	0	14	2	87.5%	100%	100%	87.5%	93.3%
Sasikumar et al. [26]	Histology or clinical/imaging FU	12	1	2	0	100%	66.7%	92.3%	100%	93.3%
Sasikumar et al. [27] *	Histology or clinical/imaging FU	5	0	1	0	100%	100%	100%	100%	100%
Verma et al. [25]	Histology	7	0	3	0	100%	100%	100%	100%	100%

Legend: Acc = diagnostic accuracy; FN = false negative; FP = false positive; FU = follow-up; NPV = negative predictive value; PPV = positive predictive value; Sen = sensitivity; Spe = specificity; TN = true negative; TP = true positive; * = study included in the systematic review but excluded from the meta-analysis for possible patient data overlap with another study of the same research group.

## Data Availability

Data supporting reported results are available in public bibliographic databases (e.g., PubMed/Medline).

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
