# Peer review of "Diagnostic Accuracy of PET/CT or PET/MRI Using PSMA-Targeting Radiopharmaceuticals in High-Grade Gliomas: A Systematic Review and a Bivariate Meta-Analysis"

_diagnostics, 2022, doi:10.3390/diagnostics12071665_

Round 1

Reviewer 1 Report

This review manuscript entitled "Diagnostic Accuracy of PET/CT or PET/MRI using PSMA-tar-2 geting Radiopharmaceuticals in High-Grade Gliomas: a Bivari-3 ate Meta-analysis" is well written and potentially helpful for HGG clinical applications.

However, PSMA target glutamate carboxypep-68 tidase II, expression level comparison is important between prostate and HGG. Perhaps a histogram would be good to have an idea to connect between this two cancers.

Author Response

Reviewer: PSMA target glutamate carboxypeptidase II expression level comparison is important between prostate and HGG. Perhaps a histogram would be good to have an idea to connect between this two cancers.

Reply: we have added comments in the discussion about PSMA expression in prostate cancers and HGG.

Reviewer 2 Report

This meta-analysis investigated the diagnostic accuracy of PSMA PET/CT or PET/MRI in brain tumors. In particular, for the differential diagnosis between high-grade glioma and low-grade glioma and detection of tumor recurrence. It has indeed been demonstrated that PSMA is overexpressed on the neovasculature of high-grade gliomas, leading to a promising new target for molecular imaging and theranostics of these types of tumors next to their application in prostate cancer.

Six studies including 157 patients in total were included resulting in a sensitivity of 98% and a specificity of 91%. For the diagnosis of HGG.

I do believe this meta-analysis has value but work on the discussion is needed to make specific conclusions or formulate hypothesis on the applications of PSMA PET in glioma imaging/theranostics.

Missing parts:

·         The introduction is quite basic. PSMA PET has been a hot topic in the last years. So more introductory information on the current status of PSMA PET should be added or at least referred to recent literature.

·         Particularly interesting for HGG is also the differentiation between tumor recurrence and radiation necrosis, since this differential diagnosis is still a hurdle in the clinic. Data on this should also be included, in the analysis or as a separate side-evaluation? Radiation necrosis typically results in endothelial cell damage and small-vessel injury and since PSMA is expressed on the neovasculature of HGG, maybe PSMA PET would be able to make a differentiation? Line 269-270 shortly mentions no uptake in radiation necrosis. Later the blood brain barrier damage is mentioned as a factor that influences the uptake of the PSMA radiopharmaceutical. Apparently, this does not influence uptake in radiation necrosis? A discussion on this is interesting to add.

·         Also interesting is the relationship between PSMA expression and tracer uptake?

·         Was only contrast enhanced MRI included for comparison? No functional MRI sequences, such as DWI and PWI? This could also be very interesting to mention and include in the analysis and should be specified.

·         What is the range of PSMA TBRs in LGG?

·         Theranostics using diagnostic+therapeutic radiolabeled PSMA radiopharmaceuticals has been studied in multiple cancer types already, see recent overview: https://pubmed.ncbi.nlm.nih.gov/34439177/. Hence the paragraph line 352-358 is not specific enough. Why is PSMA a good target for therapeutic radionuclide therapy in glioma?

Specific comments:

·         Line 117: Eligibility criteria: ‘articles not providing sufficient information to reassess the sensitivity or specificity of PSMA-targeted PET/CT or PET/MRI’. Could this be more specified?

·         Table 2. To limit the amount of words in the table, in the column ‘clinical setting’ some studies with the same clinical setting E.g. initial diagnosis of glioma (HGG vs LGG) could be pooled (merge cells)

·         Line 223: Any cerebral uptake higher than the background was interpreted as a PET positive finding. Can background be more specified? E.g. normal brain parenchyma/gray matter? No uptake of PSMA in any other brain structures? I see this was also discussed on line 320-324: the reason for no uptake in normal brain parenchyma is shortly touched: it is assumed that the PSMA radiopharmaceutical does not cross the BBB so is it assumed it can only reach the brain tumor via leaky blood vessels? Could it be that the target is on the neovasculature of GB and not on non-tumoral vasculature and can pass BBB? The word ‘abnormalities’ is maybe not ideal, rather regions with increased PSMA expression

·         Figure 2: details on how to interpret the figure could be added

·         Line 256: ‘good’ -> sufficient?

·         Line 265: distinguishing HGG from LGG?

·         Line 274-275 and throughout manuscript: at MRI/PET -> on MRI/PET

·         Line 277-281: would it be possible to add differences in TBR (GB uptake vs normal brain parenchyma) for PSMA PET vs FDG PET?

·         TBR for brain lesions can also be calculated using contralateral cerebellar uptake as background. This was not the case in these studies?

·         Point 3.5: sensitivity and specificity is quantified to detect HGG but the goal of the study was specifically the differential diagnosis between high-grade glioma and low-grade glioma and detection of tumor recurrence, which should also be quantified or specified?

·         Figure 6: top part of the table has title ‘negative LR’ but second part does not have a title ‘diagnostic odds ratio’. Also all abbreviations should be written in full in the figure legends. (FN, TN,…)

·         Line 331-336: very long sentence. Rephrase ‘fails for definitive diagnosis’

·         Line 338: rephrase `masking effect`. To be added: a downside of FDG is indeed the very high physiological uptake in normal brain, which negatively influences the contrast between tumoral tissue and normal brain tissue but FGD has also been shown to be taken up in inflammatory regions. Is this not an issue with PSMA PET?

·         Line 342-351: Which amino acid PET tracers? There are many applied in glioma. FET? I might not agree with the proposed hypothesis: Does the infiltrative component show increased amino acid transporter expression but no angiogenesis? GB is often characterized by a border with increased contrast enhancement on MRI due to leaky vessels. GB is a highly angiogenic and infiltrative tumor. Cells invade along blood vessels to support tumor growth.

·         Line 357: functional -> essential

Author Response

Reviewer: I do believe this meta-analysis has value but work on the discussion is needed to make specific conclusions or formulate hypothesis on the applications of PSMA PET in glioma imaging/theranostics.

Reply: We thank the Reviewer for the useful suggestions and comments on this manuscript. We have modified the manuscript taking into account the reviewers’comments.

Reviewer: The introduction is quite basic. PSMA PET has been a hot topic in the last years. So more introductory information on the current status of PSMA PET should be added or at least referred to recent literature.

Reply: We have added more information on the current status of PSMA PET in the introduction of the revised manuscript. In particular we have underlined that “Undoubtedly, PSMA PET is a hot topic of imaging in the last years. Recent literature data and a growing body of evidence demonstrated that PSMA targeted PET ameliorates staging and restaging of prostate cancer patients changing the management in a significant percentage of cases.” We have added recent references to this regard.

Reviewer: Particularly interesting for HGG is also the differentiation between tumor recurrence and radiation necrosis, since this differential diagnosis is still a hurdle in the clinic. Data on this should also be included, in the analysis or as a separate side-evaluation? Radiation necrosis typically results in endothelial cell damage and small-vessel injury and since PSMA is expressed on the neovasculature of HGG, maybe PSMA PET would be able to make a differentiation? Line 269-270 shortly mentions no uptake in radiation necrosis. Later the blood brain barrier damage is mentioned as a factor that influences the uptake of the PSMA radiopharmaceutical. Apparently, this does not influence uptake in radiation necrosis? A discussion on this is interesting to add.

Reply: We have added in the revised manuscript the results of the subgroup analysis about the performance of PSMA PET in suspicious HGG recurrence after treatment. “For diagnosis of HGG recurrence after treatment the pooled sensitivity and specificity of PSMA-targeted PET was 98.5% (95%CI: 90–99.8%) and 77.5% (95%CI: 55–97.3%), respectively. We have also added some comments in the discussion of the revised manuscript about the differentiation between tumor recurrence and radiation necrosis by using PSMA-targeted PET.

Reviewer: also interesting is the relationship between PSMA expression and tracer uptake

Reply: We have added more details about in vivo and in vitro PSMA expression in the results and discussion of the revised manuscript. “Only one study has correlated the in vivo and in vitro PSMA expression in gliomas by using PSMA-targeted PET and immunohistochemistry staining for PSMA, respectively. The expression of PSMA in the resected glioma tissues was confirmed to be positive in some of HGG tumors and no or only low PSMA expression for LGG has been reported." “Whereas HGG showed in vivo and in vitro high expression of PSMA, tumor samples in LGG showed no or only low PSMA-expression, thus confirming the specificity of PSMA-targeting radiopharmaceuticals for HGG detection.”

Reviewer: Was only contrast enhanced MRI included for comparison? No functional MRI sequences, such as DWI and PWI? This could also be very interesting to mention and include in the analysis and should be specified.

Reply: We have added in the revised manuscript that “contrast-enhanced MRI was used for comparison in all the included studies. Only four studies reported the use of functional MRI sequences.” Unfortunately, the authors of the included articles did not provide sufficient data about results of MRI using different sequences. Therefore, we cannot perform an analysis of PET findings correlated to different MRI sequences.

Reviewer: What is the range of PSMA TBRs in LGG?

Reply: We have reported in the revised manuscript that average TBR in HGG ranged from 11.52 to 13.95; average TBR in LGG ranged from 1.29 to 3.42.

Reviewer: Theranostics using diagnostic+therapeutic radiolabeled PSMA radiopharmaceuticals has been studied in multiple cancer types already, see recent overview: https://pubmed.ncbi.nlm.nih.gov/34439177/. Hence the paragraph line 352-358 is not specific enough. Why is PSMA a good target for therapeutic radionuclide therapy in glioma?

Reply: We have added more details in the discussion of the revised manuscript about theranostics in high grade gliomas using PSMA-targeted radiopharmaceuticals.

Reviewer; Line 117: Eligibility criteria: ‘articles not providing sufficient information to reassess the sensitivity or specificity of PSMA-targeted PET/CT or PET/MRI’. Could this be more specified?

Reply: We have added “when true positive, false positive, true negative and false negative findings were not reported”.

Reviewer: Table 2. To limit the amount of words in the table, in the column ‘clinical setting’ some studies with the same clinical setting E.g. initial diagnosis of glioma (HGG vs LGG) could be pooled (merge cells)

Reply: We have reduced the words in the column “clinical setting” of Table 2

Reviewer: Line 223: Any cerebral uptake higher than the background was interpreted as a PET positive finding. Can background be more specified? E.g. normal brain parenchyma/gray matter? No uptake of PSMA in any other brain structures? I see this was also discussed on line 320-324: the reason for no uptake in normal brain parenchyma is shortly touched: it is assumed that the PSMA radiopharmaceutical does not cross the BBB so is it assumed it can only reach the brain tumor via leaky blood vessels? Could it be that the target is on the neovasculature of GB and not on non-tumoral vasculature and can pass BBB? The word ‘abnormalities’ is maybe not ideal, rather regions with increased PSMA expression.

Reply: we have modified the statement on the background. Furthermore, we have modified the comment in the discussion about cerebral radiopharmaceutical uptake taking into account the comments of the reviewer.

Reviewer: Figure 2: details on how to interpret the figure could be added.

Reply: we have added details on how interpret the figure 2.

Reviewer: Line 256: ‘good’ -> sufficient?

Reply: we have modified “good” in “adequate”.

Reviewer: Line 265: distinguishing HGG from LGG?

Reply: We have reported “distinguishing HGG from LGG” in the revised manuscript.

Reviewer: Line 274-275 and throughout manuscript: at MRI/PET -> on MRI/PET

Reply: We have reported “on MRI/PET” in the revised manuscript

Reviewer: Line 277-281: would it be possible to add differences in TBR (GB uptake vs normal brain parenchyma) for PSMA PET vs FDG PET?

Reply: We have added differences in TBR for PSMA PET vs FDG PET.

Reviewer: TBR for brain lesions can also be calculated using contralateral cerebellar uptake as background. This was not the case in these studies?

Reply: We have added that the included studies defined background uptake as contralateral cerebral uptake in some studies or contralateral cerebellar uptake in other studies.

Reviewer: Point 3.5: sensitivity and specificity is quantified to detect HGG but the goal of the study was specifically the differential diagnosis between high-grade glioma and low-grade glioma and detection of tumor recurrence, which should also be quantified or specified?

Reply: we have added subgroup analyses results in the different settings.

Reviewer: Figure 6: top part of the table has title ‘negative LR’ but second part does not have a title ‘diagnostic odds ratio’. Also all abbreviations should be written in full in the figure legends. (FN, TN,…)

Reply: we have modified figures according to the reviewer’s comments.

Reviewer: Line 331-336: very long sentence. Rephrase ‘fails for definitive diagnosis’

Reply: We have rephrased this statement.

Reviewer: Line 338: rephrase `masking effect`. To be added: a downside of FDG is indeed the very high physiological uptake in normal brain, which negatively influences the contrast between tumoral tissue and normal brain tissue but FDG has also been shown to be taken up in inflammatory regions. Is this not an issue with PSMA PET?

Reply: we have rephrased the statement and added the suggestions of the reviewer.

Line 342-351: Which amino acid PET tracers? There are many applied in glioma. FET? I might not agree with the proposed hypothesis: Does the infiltrative component show increased amino acid transporter expression but no angiogenesis? GB is often characterized by a border with increased contrast enhancement on MRI due to leaky vessels. GB is a highly angiogenic and infiltrative tumor. Cells invade along blood vessels to support tumor growth.

Reply: we have rephrased this statement.

Reviewer:Line 357: functional -> essential

Reply: we have reported “essential” as suggested.

Reviewer 3 Report

Very nice and indeed important study with well-presented and explained results. I just have a few remarks: 

Line 67: Why is it important to emphasize that reference 7 (self citation) is evidence-based? 

Line 116: Typo: Positron emitters OR positron emitting radionulides 

Line 306: Can the authors comment on a possible data bias due to the fact that two different tracers were used? What differences exist in the uptake/biodistribution patterns of PSMA-11 and PSMA-617? 

Line 365: Could the authors comment on concerns about radiotoxicity of brain tissue, dose limiting structures and crossfire effects of ß emitters? 

Author Response

Reviewer: Line 67: Why is it important to emphasize that reference 7 (self citation) is evidence-based?

Reply: we modified the statement in the revised manuscript.

Reviewer: Line 116: Typo: Positron emitters OR positron emitting radionuclides

Reply: We have modified the statement according to the reviewer’s suggestion.

Reviewer: Line 306: Can the authors comment on a possible data bias due to the fact that two different tracers were used? What differences exist in the uptake/biodistribution patterns of PSMA-11 and PSMA-617?

Reply: We have added a comment in the discussion of the revised manuscript about the different PSMA-targeting radiopharmaceuticals.

Reviewer: Line 365: Could the authors comment on concerns about radiotoxicity of brain tissue, dose limiting structures and crossfire effects of ß emitters?

Reply: We have added a statement in the discussion underlining that for a theranostic approach in HGG by using PSMA-targeted therapy, concerns about radiotoxicity of brain tissue, dose limiting structures and crossfire effects of beta-emitters should be better evaluated.

Reviewer 4 Report

This paper evaluates the specificity and sensitivity of PET imaging using the PSMA-targeting radiotracers in determining high-grade glioma. PSMA PET imaging is one of the hottest topics in PET imaging research, so a systematic paper review of PSMA imaging might be required. However, the reviewer raises the following concerns about this review paper.

1. As the authors mentioned in the manuscript in the discussion part, this paper indicated that this paper considered only six papers for the meta-analysis. This paper looks to satisfy the minimum requirement for a successful meta-analysis. The authors also performed a comprehensive search to include the PSMA PET studies for high-grade glioma in the current review paper. Despite their efforts, they could find only six papers to meet their standards. Therefore, the reviewer concluded that this kind of study is too early to make conclusions. 

2. Please explain why the PET study using [68Ga]PSMA-617 is important. Because the rest of the literature used [68Ga]PSMA-11, the comparison of the papers using the same radiotracer might increase confidence.

3. In the introduction, please argue why PSMA imaging can be better for the diagnosis of high-grade glioma than other PET imaging such as [18F]FDOPA, [11C]methionine, and [18F]FET.

4. If available, can you compare the current meta-analysis with the similar meta-analysis studies using other PET imaging such as [18F]FDOPA, [11C]methionine, and [18F]FET in the discussion part?

5. The reviewer asks to fix the following errors in the manuscript.

Line 52: High-grade glioma can be abbreviated to HGG.

Line 127: screneed --> screened

4. In the 

Author Response

Reviewer: As the authors mentioned in the manuscript in the discussion part, this paper indicated that this paper considered only six papers for the meta-analysis. This paper looks to satisfy the minimum requirement for a successful meta-analysis. The authors also performed a comprehensive search to include the PSMA PET studies for high-grade glioma in the current review paper. Despite their efforts, they could find only six papers to meet their standards. Therefore, the reviewer concluded that this kind of study is too early to make conclusions.

Reply: we thank the reviewer for the useful comments. We did not know the number of eligible articles before performing the comprehensive literature search, but we had recognized that this was a hot topic and it deserved a systematic review. The number of included articles were eight for the systematic review and six for the meta-analyses. Taking into account the reviewer’s comment, we have added in the discussion of the revised manuscript and recognized that the limited number of included articles and other limitations hampered to obtain definitive conclusions on this topic. However, definitive conclusions are not mandatory in a systematic review or meta-analysis and the conclusions of these evidence-based articles are related to the included original articles (unfortunately, most of the published systematic reviews and meta-analyses cannot provide definitive conclusions). We believe that the main aim of this work was to answer to the review question through an appropriate methodology (and we thank the reviewer for having appreciated our efforts in this regard) and to suggest further studies to obtain definitive conclusions ("what is missing in the literature"). Taking into account the provided results, we believe that we could conclude that the diagnostic performance of PSMA-targeted PET in HGG seems promising even if further well-designed original studies are needed.

Reviewer: Please explain why the PET study using [68Ga]PSMA-617 is important. Because the rest of the literature used [68Ga]PSMA-11, the comparison of the papers using the same radiotracer might increase confidence.

Reply: we have added a comment in the discussion about different PSMA-targeted radiopharmaceuticals used. “Most of the included studies used [68Ga]Ga-PSMA-11 as PSMA-targeting radiopharmaceutical [20,21,23-27], whereas one study only used [68Ga]Ga-PSMA-617 [22] and this can be a potential bias in the meta-analysis as the comparison of studies using the same PSMA-targeting radiopharmaceutical might increase confidence. However, even if PSMA-617 has been reported to have slower tumor accumulation and clearance kinetics than PSMA-11 [34], a significant difference in the detection of HGG using different PSMA-targeting radiopharmaceuticals was not demonstrated. Studies comparing different PSMA-targeting radiopharmaceuticals in detecting HGG are warranted.”

Reviewer: In the introduction, please argue why PSMA imaging can be better for the diagnosis of high-grade glioma than other PET imaging such as [18F]FDOPA, [11C]methionine, and [18F]FET.

Reply: We have added in the introduction of the revised manuscript the following statement “Compared to other PET radiopharmaceuticals used for HGG, PSMA-targeting radiopharmaceuticals may evaluate a different tumor characteristic (tumor neoangiogenesis). Interestingly, for HGG there could be also a potential advantage of a theranostic approach offered by PSMA-targeted agents which is not possible with [18F]FDG and radiolabelled amino acid tracers.

Reviewer: If available, can you compare the current meta-analysis with the similar meta-analysis studies using other PET imaging such as [18F]FDOPA, [11C]methionine, and [18F]FET in the discussion part?

Reply: we have added in the discussion more information on other meta-analyses using [18F]FDOPA, [11C]methionine, and [18F]FET, according to the reviewer’s suggestion.

Reviewer: The reviewer asks to fix the following errors in the manuscript.

Line 52: High-grade glioma can be abbreviated to HGG.

Line 127: screneed --> screened

Reply: We thank the reviewer. We have checked and corrected the errors.

Round 2

Reviewer 2 Report

All comments were addressed. Please take note of a wrong sentence structure on line 430 and line 450. The discussion improved significantly with inclusion of open question like radiation necrosis, comparison with other glioma PET radiopharmaceuticals, theranostics,...

Author Response

We have modified the sentence structure as suggested by the reviewer.

Reviewer 4 Report

Generally, the authors fulfilled most of the reviewer's requests.

They also made clear the limitation of this review article.

However, as the authors point out, it looks very early to discuss the sensitivity and specificity using the systematic analysis despite the PSMA PET imaging is a very hot topic in this field.

The work can be done again in the future for better accurate analysis when PSMA PET imaging becomes a prevalent imaging modality.

Author Response

We thank the reviewer for having appreciated our efforts in ameliorating the quality of this manuscript.

Taking into account the comment of the reviewer, we have added this statement in the discussion: "overall, we would like to suggest to update this evidence-based analysis in the future when more original articles on the selected topic will be published to obtain a more accurate analysis of the diagnostic performance of the index test"